# Immune Evasion of SARS-CoV-2 Omicron Subvariants

**DOI:** 10.3390/vaccines10091545

**Published:** 2022-09-16

**Authors:** Hanzhong Ke, Matthew R. Chang, Wayne A. Marasco

**Affiliations:** 1Department of Cancer Immunology and Virology, Dana-Farber Cancer Institute, Harvard Medical School, 450 Brookline Avenue, Boston, MA 02215, USA; 2Department of Medicine, Harvard Medical School, Boston, MA 02215, USA

**Keywords:** SARS-CoV-2, Omicron, Omicron subvariants, immune evasion, spike mutation, molecular basis

## Abstract

Since the SARS-CoV-2 Omicron variant (B.1.1.529) was declared a variant of concern (VOC) by the WHO on 24 November 2021, it has caused another global surge of cases. With extensive mutations in its spike glycoprotein, Omicron gained substantial capabilities to evade the antiviral immunity provided by vaccination, hybrid immunity, or monoclonal antibodies. The Omicron subvariants BA.1, BA.2, BA.2.12.1, BA.4 and BA.5 extended this immune evasion capability by having additional unique mutations in their respective spike proteins. The ongoing Omicron wave and emergence of new Omicron subvariants leads to additional concerns regarding the efficacy of the current antiviral measurements. To have a better understanding of the Omicron subvariants, this review summarizes reports of the immune evasion of subvariants BA.1, BA.2, BA.2.12.1, BA.4, and BA.5 as well as the molecular basis of immune evasion.

## 1. Introduction

Since emerging in Wuhan in late 2019, the severe acute respiratory syndrome coronavirus 2 (SARS-CoV-2) has been estimated to have infected over a half billion people and to have caused over 6 million deaths (Accessed on 3 August 2022 https://covid19.who.int). As a single-stranded positive sense RNA virus, SARS-CoV-2 has been continousouly evolving through mutations due to its error-prone RNA polymerase [1] and anti-spike immune pressure, resulting in successive waves of infection by these mutated strains. Among those, the WHO has designated Alpha (B.1.1.7), Beta (B.1.351), Gamma (P.1), Delta (B.1.617.2) and Omicron (B.1.529) as variants of concern (VOCs), as these strains were responsible for the majority of cases during their respective time periods. Omicron was first reported in South Africa and Botswana [2] then rapidly spread around the world. With vaccination and other antiviral measurements, Alpha, Beta, Gamma, and Delta have been successfully contained. However, the Omicron variant continues to spread with unprecedented transmissibility, demonstrating remarkable immune evasion and the ability to infect both vaccinated and convalescent individuals. Omicron has further developed several subvariants including BA.1, BA.2, BA.2.12.1, BA.4, and BA.5, which contain both shared and unique mutations. These Omicron subvariants extend the ongoing pandemic and display further immune evasion capabilities.

The infection of SARS-CoV-2 primarily depends on the interaction of its spike protein and host cell receptor ACE2 [3,4,5]. The SARS-CoV-2 spike protein is a class I fusion transmembrane glycoprotein composed of S1 and S2 subunits. The S1 subunit contains an N-terminal domain (NTD) with an unclear function, and a receptor-binding domain (RBD) that binds to ACE2 (Figure 1). The S2 subunit includes the fusion peptide and Heptad repeats 1 and 2, and it directly mediates viral fusion and host entry after cleavage by host proteases such as TMPRSS2 and furin [3]. Omicron’s spike protein contains > 30 mutations, including at least 15 in the RBD, which is the principal target for neutralizing antibodies [6]. The NTD is also a neutralizing target by antibodies that bind to its so-called antigenic supersite involving the N3 (residues 141 to 156) and N5 (residues 246–260) loops [7,8]. Omicron escapes NTD-directed neutralization through extensive mutations in this region as well. Note that SARS-CoV-2 Omicron subvariants also have extensive mutations in the non-spike protein region, including nonstructural proteins (nsp), non-spike structural proteins, and accessory proteins. These non-spike proteins play critical roles in virus replication, fitness, host immune evasion, and transmission. For example, SARS-CoV-2 has 16 nsps and among them, nsp1, nsp3, nsp4, nsp5, nsp6, nsp13, and nsp14 facilitate virus replication, and nsp1, nsp3, nsp6, and nsp13 suppress host innate immune response for virus survival. Other structural proteins, including E protein, N protein, and M protein support virus replication, while N protein and M protein also inhibit the host innate immune response to support virus immune evasion. As for the accessory proteins, ORF3a, ORF3b, ORF6, ORF7a, ORF7b, ORF8, and ORF9b interfere with host innate immune responses including interferon (IFN) signaling and IFN-stimulated gene (ISG) production, as well as interleukin signaling. Omicron-specific mutations in these non-spike regions will certainly pose impacts on viral pathogenesis and host immune evasion. These impacts have been comprehensively reviewed by A. Hossain et al. [9].

In this opinion article, with a specific focus on the spike protein, we summarize the immune evasion from either vaccine-elicited or monoclonal antibodies of Omicron subvariants including BA.1, BA.2, BA.2.12.1, BA.4, and BA.5, which are currently the major circulating strains. The molecular basis of the immune evasion is also discussed. Since not all mutations favor the virus in terms of transmission or immune evasion, this opinion article will only focus on those mutations that benefit the virus.

## 2. BA.1 Subvariant

Omicron BA.1 is the original Omicron subvariant first identified in November 2021 when the CoV-19 vaccines were widely administered and therapeutic antibodies were also available. It is highly transmissible and outcompeted Delta within weeks to become the dominant circulating strain [10]. These facts clearly demonstrate that Omicron BA.1 can escape the immune protections offered by both vaccines and antibodies. Multiple reports comparing the Omicron variant (BA.1) to the ancestral strains (Wuhan, Alpha), demonstrated that Omicron has higher escape capability from convalescent and vaccine-elicited neutralization [11,12,13]. However, BA.1 seems to have no superiority in evading neutralization from the hybrid immunity elicited by the combination of vaccination and infection with ancestral strains [11]. Booster vaccinations demonstrated greater efficiency than two-dose vaccinations [14], but BA.1 still showed a higher capacity for immune evasion than Delta [12,14,15]. With the exception of sotrovimab, many therapeutic monoclonal antibodies (casirivimab, imdevimab, bamlanivimab, etesevimab) exhibited significantly reduced neutralizing capabilities against BA.1 [14,15].

Compared to ancestral Wuhan-Hu-1, BA.1 contains 34 amino acid substitutions, deletions, and insertions in the spike [16,17]. There are 15 mutations in the RBD that lead to the evasion of RBD-targeted antibodies, including G339D, S371L, S373P, S375F, K417N, N440K, G446S, S477N, T478K, E484A, Q493R, G496S, Q498R, N501Y, Y505H (Figure 1) [16,18]. Based on binding epitopes, RBD-targeting antibodies are categorized into six classes (I–VI) [16,18]. K417N, Q493R, N501Y, and Y505H mutations dampen the binding of class I antibodies, S477N, T478K, E484A, Q493R, and Y505H impair class II binding, while E484A and Q493R impair class I and II binding [16,18,19]. Alterations in the NTD include A67V, del69-70, T95I, G142D, del143-145, N211I, del212, and ins214EPE [16,18]. A67V together with del69-70 changes the conformation of the N2 loop. N211I, del212, and ins214EPE alter the configuration of the N4 loop. G142D and del143-145 leads to a reconfiguration of the N3 loop. These reorganizations together change the NTD sites, resulting in the evasion of NTD-targeting antibodies [16,18].

## 3. BA.2 Subvariant

Omicron BA.2 arose soon after BA.1 and with higher levels of transmissibility, quickly outcompeted BA.1 [20,21,22]. As of May 2022, BA.2 has shown global dominance, though the COVID-19 vaccine effectiveness against BA.2 is not further reduced compared to BA.1 [23]. Neutralization assays and antigenic cartography analysis have revealed that BA.1 and BA.2 are antigenically distinct [24,25]. Like BA.1, BA.2 escapes the immune protection elicited by vaccination, previous infection, and hybrid immunity [26,27]. As for antibody evasion, mutations in BA.2 enable the evasion of S309 (sotrovimab), which is a distinct difference when compared to BA.1 [28].

BA.1 and BA.2 share 21 alterations, while BA.1 contains 13 unique mutations and BA.2 contains 8 [27,28]. These notable differential mutation profiles may explain their distinct antigenic characteristics, and the mechanism of BA.2’s immune evasion is not exactly the same as BA.1. Extensive but different mutations occurred in the NTD of BA.2, which resulted the disruption of the NTD loops to prevent antibody binding **(**Figure 1) [18,29]. The shared mutations in the RBD allow BA.2 to escape class I and II antibodies as BA.1 does, while the unique mutations give BA.2 additional power to escape more neutralizing antibodies. For example, because BA.2 does not have the G446S mutant, this leads BA.2 to be more sensitive to class V antibodies than BA.1 (Figure 1) [29]. These findings are in line with its superiority over BA.1 in terms infectivity and rate of spread.

## 4. BA.2.12.1 Subvariant

The BA.2.12.1 subvariant is one of descendants from BA.2, and compared to BA.2 it has two more mutations in the spike, L452Q and S704L (Figure 1). It emerged in the northeast regions of the US in early February, 2022 and reports showed that BA.2.12.1 is more resistant to sera from vaccinated individuals than BA.2 [30,31,32]. Though neutralizing titers against BA.2.12.1 decreased significantly in donors with three mRNA doses and a previous infection with BA.1 and BA.2, those infected with BA.1 exhibited a greater degree of immune evasion [31,32,33,34]. All this evidence clearly shows that the BA.2.12.1 subvariant further extends immune evasion, rendering current vaccine regiments less effective. Studies have demonstrated that L452Q substitution causes a 2–5-fold resistance of monoclonal antibodies Bebtelovimab and Cilgavimab [35]. The mutation of L452 to either Q452 (BA.2.12.1) or R452 (BA.4/5) causes steric hindrance, which impacts the binding of class II and class III antibodies in general [30]. However, the role of the S704L mutation is not clear.

## 5. BA.4/5 Subvariants

BA.4 and BA.5 are the newest members of Omicron’s growing family, which initially emerged in South Africa and are now spreading globally [36]. Because BA.4 and BA.5 have identical spike sequences, they will be referred to as BA.4/5 hereafter. BA. 4 and BA.5 share unique mutations, L452R and F486V (Figure 1). BA.4/5 is more resistant than BA.2.12.1, resulting in more breakthrough infections [30,32,33,34]. For example, while BA.2.12.1 is 1.8-fold more resistant to the sera of triple-vaccinated individuals than BA.2, and BA.4/5 is 4.5-fold more resistant [30]. Another report showed that BA.2.12.1 has a 2.2-fold lower capability of neutralizing titers compared to BA.1, and BA.4/5 has 3.3-fold lower titers [34]. While L452R mimics the L452 substitution in BA.2.12.1, F486V further facilitates the escape from class I and II antibodies, including 1–20, REGN10933, and Ly-CoV-555, likely due to the similar steric hindrance that is caused by the phenylalanine-to-valine substitution [30,37]. It is worthy to note that the F486V mutation of BA.4/5 reduces the binding affinity of the RBD to ACE2 due to the loss of the hydrophobic interaction [30]. However, the reverse mutation of arginine R493 to the wild-type glutamine Q493 restores the binding affinity to ACE2 and rescues the viral fitness of the BA.4/5 subvariant [30]. These studies suggest that antibodies triggered by vaccinations are less effective at blocking the BA.4/5 subvariant.

## 6. Conclusion Remarks

SARS-CoV-2 keeps evolving to escape the “walls” that were built by vaccination, previous infection, and their combinations. Additionally, the Omicron variant continues to break through adaptive immune responses by the development of further subvariants. With an established spectrum of mutations, the Omicron subvariants are able to evade previous immunity and antiviral measurements including therapeutic monoclonal antibodies while rendering the current vaccines less effective. As the plethora of Omicron subvariants continue to grow, the development of new generations of vaccines and monoclonal antibodies is a pressing need. For example, the inclusion of Omicron-subvariant-specific vaccines in our armamentarium holds promise for preventing breakthrough infections. In this regard, the US Food & Drug Administration (FDA) has authorized Moderna, Pfizer-BioNTech bivalent COVID-19 vaccines as a booster dose to improve the protection against Omicron subvariants. Both bivalent COVID-19 vaccines contain a combination of the original and the Omicron BA.4/5 spike, and superior efficacy has been shown in these vaccine formulations (https://www.fda.gov Accessed on 3 August 2022). Nasal vaccines are exceptional at eliciting respiratory mucosal immunity, and showed great potential for Omicron protection in a mice model [38,39]. In addition to mRNA and viral-based vaccine platforms, protein-based adjuvanted vaccines (Novavax) that broaden the current immune responses are also a valuable approach to contain Wuhan and future Omicron subvariants [40]. Furthermore, RBD-dimeric vaccines, mosaic RBD nanoparticle vaccines, and conservative S2-targeting vaccines have shown their potential for pan-beta-coronaviruses, including SARS-CoV-2 and the SARS-like sarbecoviruses, as well as human endemic coronavirus protections by inducing broadly neutralizing antibodies [41,42,43]. These active and passive vaccination efforts collectively contribute to our long-term path to pan-beta-coronavirus, or pan-coronavirus, protection.

## Figures and Tables

**Figure 1 vaccines-10-01545-f001:**
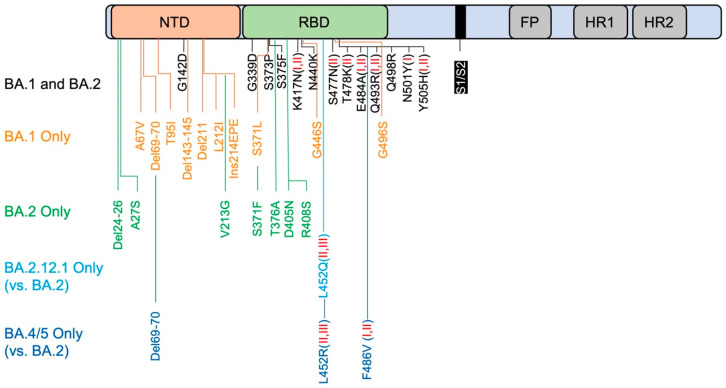
Spike mutations that contribute to immune evasion. Schematic representation of spike (not to scale) contains NTD (N-terminal domain), RBD (receptor-binding domain), S1/S2 cleavage site, FP (fusion peptide), HR1 (Heptad repeat) and HR2. Del: Deletion, Ins: Insertion. Red I and/or II in brackets indicate clearly described class I and/or II antibodies that have been dampened by the particular mutation.

## Data Availability

The study did not report any data.

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
