# Peer review of "Immune Evasion of SARS-CoV-2 Omicron Subvariants"

_vaccines, 2022, doi:10.3390/vaccines10091545_

Round 1
Reviewer 1 Report
The manuscript comprehensively describes Omicron subvariants and potential mechanism of immune evasion. The manuscript is clearly written and the data well presented. I believe that the manuscript is well timed and interesting to read.
Author Response
The manuscript comprehensively describes Omicron subvariants and potential mechanism of immune evasion. The manuscript is clearly written and the data well presented. I believe that the manuscript is well timed and interesting to read.
Response: Thank you for reading our work and leaving us such positive comments.
Reviewer 2 Report
Ke et al have performed a review of the identified SARS-CoV-2 omicron subvariants with emphasis placed on the degree of immune evasion specific to each one. The result is a concise summary of the topic that is likely to be of interest to readers of Vaccines.
The authors may wish to consider a few small revisions to the present work:
- at line 59, the date is presumably meant to be November 2021
- the information found between lines 74 and 80 is quite dense and therefore may be better suited to presentation in a table (specifically the evasion of antibody classes as the mutations themselves are already well covered by figure 1)
- since figure 1 does not directly touch on vaccine efficacy, its citation at line 110 feels slightly out of place
Author Response
Reviewer 2
Ke et al have performed a review of the identified SARS-CoV-2 omicron subvariants with emphasis placed on the degree of immune evasion specific to each one. The result is a concise summary of the topic that is likely to be of interest to readers of Vaccines.
The authors may wish to consider a few small revisions to the present work:
- at line 59, the date is presumably meant to be November 2021
Response: Thank you for picking up this error. Yes, it was meant to be Nov. 2021 and has been corrected in the revision at line 76.
- the information found between lines 74 and 80 is quite dense and therefore may be better suited to presentation in a table (specifically the evasion of antibody classes as the mutations themselves are already well covered by figure 1)
Response: Thank you for the suggestion. Because we have a similar but more comprehensive table published elsewhere (Chang et al., eBioMedicine. 2022; 80:104025), we decided to incorporate this information into revised Figure 1 by adding red I and/or II in brackets to indicate class I, II, and/or III antibodies that have been dampened by the particular mutation. The new Figure 1 and legend is provided in the revision, the reference (Chang et al., eBioMedicine. 2022; 80:104025) has been cited as ref 19 in the text at line 97.
- since figure 1 does not directly touch on vaccine efficacy, its citation at line 110 feels slightly out of place
Response: We have removed the (Figure 1) citation in the text in line 140 of the revision to accommodate this comment.
Reviewer 3 Report
Authors should include and explain the below comments
1. Neutralization escape of antibodies by BA.2.12.1, BA.4/5.
2. Receptor-binding domain (RBD) mutations
3. Non-spike proteins may also impart epistatic effects
4. The authors should include R493Q reversion mutation along with L452R and F486V
5. Any alternative design or strategies in fighting against these variants
6. What are the differences between these variants and their progeny sub-variants
7. Mutations such as NSP3, NSP6, NSP13, M protein, ORF7b, and ORF9b explain briefly
Author Response
Authors should include and explain the below comments
- Neutralization escape of antibodies by BA.2.12.1, BA.4/5.
Response: Yes, we have discussed the antibodies and mechanisms that have been escaped by BA.2.12.1 L452Q mutations by writing in lines 142 – 144: “Studies have demonstrated that L452Q substitution causes 2-5 fold resistance of mono-clonal antibodies Bebtelovimab and Cilgavimab(35). The mutation on L452 to either Q452 (BA.2.12.1) or R452 (BA.4/5) causes steric hindrance which impacts the binding of class II and class III antibodies in general(30).”
We also discussed the antibody and evasion mechanism of BA.4/5 in lines 154 – 156: “While L452R mimics the L452 substitution in BA.2.12.1, F486V further facilitates escaping from class I and II antibodies, including 1-20, REGN10933, Ly-CoV-555, likely due to the similar steric hindrance that caused by Phenylalanine-to-Valine substitution”. A new reference (Tuekprakhon et al., Cell. 2022; 185(14):2422-33. e13) has been cited as ref 37 in line 156.
- Receptor-binding domain (RBD) mutations
Response: Yes, we have indicated the receptor-binding domain (RBD) mutations in Figure 1. And we highlighted the mutations in the RBD regions which is the major target for neutralizing antibodies in lines 50-51 of the revision. We then discussed the RBD mutations of each variant in its respective section.
- Non-spike proteins may also impart epistatic effects
Response: We agree that non-spike proteins are playing critical roles in virus pathogenesis and host immune evasions. Any mutations or Omicron-specific mutations in these regions may impact their authentic functions. To address this, we added texts in lines 54 – 68 in the revision to highlight the nonstructural proteins (nsps), structural proteins (excluding spike), and accessory proteins by saying “Note that SARS-CoV-2 Omicron subvariants also have extensive mutations in non-spike protein region, including nonstructural proteins (nsps), structural proteins (except spike), and accessory proteins. These non-spike proteins play critical roles in virus replication, fitness, host immune evasion, and transmission. For example, SARS-CoV-2 has 16 nsps and among them, nsp1, nsp3, nsp4, nsp5, nsp6, nsp13, and nsp14 facilitates virus rep-lication, nsp1, nsp3, nsp6, and nsp13 suppress host innate immune response for virus survival. Other structural proteins, including E protein, N protein, and M protein sup-ports virus replication, while N protein and M protein also inhibits host innate immune response to support virus immune evasion. As for accessory proteins, ORF3a, ORF3b, ORF6, ORF7a, ORF7b, ORF8, and ORF9b interfere with host innate immune responses including interferon (IFN) signaling and IFN-stimulated gene (ISG) production, as well as interleukin signaling. Omicron-specific mutations in these non-spike regions will cer-tainly pose impacts on viral pathogenesis and host immune evasion. These impacts have been comprehensively reviewed by A. Hossain et al. (9).” Because this part has been reviewed comprehensively by A. Hossain et al., (Hossain, et al., Microb Patho. 2022; 170:105699), we cited this reference in the text (as ref 9 in line 68) and referred audiences to this paper.
- The authors should include R493Q reversion mutation along with L452R and F486V
Response: In lines 156 – 177, the R493Q has been discussed by saying “It is worthy to note that F486V mutation of BA.4/5 reduces the binding affinity of RBD to ACE2 due to loss of hydrophobic interaction(30). However, the reverse mutation of Ar-ginine, R493 to wild-type Glutamine Q493 restores the binding affinity to ACE2, com-pensating the viral fitness of BA.4/5 subvariant(30).”
- Any alternative design or strategies in fighting against these variants
Response: To address this comment, we have discussed newly FDA Approved Moderna and Pfizer-BioNTech COVID-19 bivalent vaccine as booster dose, protein based adjuvanted vaccines, RBD-dimeric vaccines, mosaic RBD nanoparticle vaccines, and conservative S2-targeting vaccines in lines 188 – 202 of the revision by saying “In this regard, US Food & Drug Administration (FDA) has authorized Moderna, Pfiz-er-BioNTech bivalent CoVID-19 vaccines as a booster dose to improve the protection against Omicron subvariants. Both bivalent COVID-19 vaccines contain original and Omicron BA.4/5 spike and superior efficacy has been shown in these vaccine formulations (https://www.fda.gov). Nasal vaccines are exceptional at eliciting respiratory mucosal immunity, has shown great potential of Omicron protection in mice model (38, 39). In addition to mRNA and viral based vaccine platforms, protein based adjuvanted vaccines (Novavax) that broaden current immune responses are also valuable approach to contain current and future Omicron subvariants (40). Furthermore, RBD-dimeric vaccines, mosaic RBD nanoparticle vaccines, and conservative S2-targeting vaccines have shown their potential of pan-beta-coronavirus including SARS-CoV-2, SARS-like sarbecoviruses, and human endemic coronavirus protections by inducing broadly neutralizing antibodies (41-43). These active and passive vaccination efforts collectively contribute to our long-term path to pan-beta-coronavirus, or pan-coronavirus protection.” New ref citations 38, 39, 41, 42, and 43 have been added in lines 194 and 200, respectively.
- What are the differences between these variants and their progeny sub-variants
Response: In each of the Omicron subvariants, BA.1, BA.2, BA.2.12.1, and BA.4/5 that have been discussed, we have introduced their origins in the text and defined their unique mutations in Figure 1. However, because their progeny sub-variants have not established substantial infection cases or drew significant concerns or both, please excuse us for not covering them in this manuscript.
- Mutations such as NSP3, NSP6, NSP13, M protein, ORF7b, and ORF9b explain briefly
Response: Similar to our response to 3, we acknowledge the critical roles of non-spike proteins including NSP3, NSP6, NSP13, M protein, ORF7b, and ORF9b. We have addressed in lines 54 – 68 in the revision and referred audience to A. Hossain et al., (Hossain, et al., Microb Patho. 2022; 170:105699).
Round 2
Reviewer 3 Report
Accept in present form